# Bus Stops Near Schools Advertising Junk Food and Sugary Drinks

**DOI:** 10.3390/nu12041192

**Published:** 2020-04-24

**Authors:** Donna Huang, Amanda Brien, Lima Omari, Angela Culpin, Melody Smith, Victoria Egli

**Affiliations:** 1School of Medicine, Faculty of Medical and Health Sciences, The University of Auckland, Auckland 1023, New Zealand; duha742@aucklanduni.ac.nz; 2Healthy Auckland Together, Auckland Regional Public Health Service, Auckland 1155, New Zealand; abrien@adhb.govt.nz (A.B.); aculpin@adhb.govt.nz (A.C.); 3School of Population Health, Faculty of Medical and Health Sciences, The University of Auckland, Auckland 1023, New Zealand; lima_omari@windowslive.com; 4The School of Nursing, Faculty of Medical and Health Sciences, The University of Auckland, Auckland 1023, New Zealand; melody.smith@auckland.ac.nz

**Keywords:** food and beverage marketing, children, neighbourhoods, built environment

## Abstract

Children rarely understand the full extent of the persuasive purpose of advertising on their eating behaviours. Addressing the obesogenic environments in which children live, through a quantification of outdoor advertising, is essential in informing policy changes and enforcing stricter regulations. This research explores the proportion of bus stop advertisements promoting non-core food and beverages within walking distance (500 m) from schools in Auckland, New Zealand while using Google Street View. Information was collected on: school type, decile, address, Walk Score^®^, and Transit Score for all 573 schools in the Auckland region. Ground-truthing was conducted on 10% of schools and showed an alignment of 87.8%. The majority of advertisements on bus shelters were for non-food items or services (n = 541, 64.3%). Of the advertisements that were for food and/or beverages, the majority were for non-core foods (n = 108, 50.2%). There was no statistically significant difference between the variables core and non-core food and beverages and School decile (tertiles), Walk Score (quintiles), and Transit Score (quintiles). 12.8% of all bus stop advertisements in this study promoted non-core dietary options; highlighting an opportunity for implementing stricter regulations and policies preventing advertising unhealthy food and drink to children in New Zealand.

## 1. Introduction

The prevalence of children affected by overweight or obesity has increased from 4% to 18% in the period from 1975 to 2016 [1]. Obesity is a preventable risk factor for disease [1], with a complex plethora of contributing factors from individual factors, including behaviour and genetics, to environmental factors, such as food environments. The results of individual-level interventions targeting health behaviours have proven to be unsuccessful when compared to changes in public policies and wider environmental determinants [2,3,4]. The importance of health promotion strategies targeted to the level of the environment is evidenced through the inability of individual behavioural approaches to reduce paediatric obesity rates [2,5].

Overweight and obesity produces ongoing financial and health costs for the individual and the health system [6]. Short-term consequences that are associated with obesity in childhood include psychological ill health, increased cardiovascular risk, and asthma [7]. Additionally, the persistence of obesity into adulthood and related comorbidities may arise in the long term, such as increased risk of cancer and other chronic diseases [7,8].

Many inequities regarding obesity prevalence among children exist. In Aotearoa, New Zealand (hereafter, NZ), and around the world, people who live in the most socioeconomically deprived neighbourhoods are more likely to be affected by excess body size [9,10]. When compared to children living in the least socioeconomically deprived areas, children living in the most deprived areas were 1.6 times as likely to be obese [11]. There is lower access to healthy foods [12] and greater exposure to unhealthy foods [12,13,14,15] in more deprived neighbourhoods. In NZ, Māori (New Zealand’s Indigenous population) are more likely to live in the most deprived areas (n = 140,886, 23.5%) as compared to Non-Māori (n = 232,779, 6.8%) [11]. Females (n = 75,117, 24.3%) are also more likely to live in the most deprived areas when compared to males (n = 65,712, 22.8%) [11]. Children of Māori (28.4%) and Pacific Island (15.5%) ethnicity have considerably higher prevalence of obesity compared to New Zealand European/Other ethnicities (8.2%) [9].

According to the United Nations, all children have the right to a healthy food environment, and governments should prohibit all forms of advertising that are likely to be visible to children of unhealthy foods and beverages [16]. NZ ratified the Convention on the Rights of the Child in 1993 [17]; however, most of the food advertising seen by children today continues to promotes unhealthy products [18]. This does not align with the Convention on the Rights of the Child [19]. The presence of unhealthy food and drink advertisements in areas children visit every day impacts upon children’s food choices [20,21]; this is highlighted in the UNICEF-Innocenti framework on food systems for children and adolescents [22]. Outside of the family setting, school is the most important environment for children and adolescents [23,24,25]. This research is underpinned by the socio-ecological model of health behaviours [26].

Studies show that children rarely understand the full extent of the persuasive purpose of advertising on their eating behaviours [27,28,29]. Adults find it difficult to make healthy food choices when exposed to outdoor advertising [30]; thus, it is imperative children are not also expected to be able to do so [16]. With the rate of change of child obesity stagnating recently in NZ [9], different approaches need to be explored to reduce the prevalence and prevent associated harms. Although supporting behavioural changes at the individual level is important, addressing the obesogenic environments in which people live is essential and can only be sufficiently achieved by policy changes [31,32,33]. A greater understanding of the extent to which children are exposed to advertising in their neighbourhoods and around their schools is needed in order to inform advertising policies in NZ. Policies monitoring and restricting marketing to children require reformation because children are more likely to choose advertised products [34]. Currently, the industry in NZ is self-regulated [35], which is not sufficient to protect children [16,36].

This study explores the proportion of bus stop advertisements promoting non-core (not recommended to be marketed to children based on the WHO Regional Office for Europe Nutrient Profile Model) [18,37] food and beverages within walking distance (500 m) from schools in Auckland, NZ. Children pass bus stops while walking, cycling, bussing, or being driven to school. The advertisements on bus stops are large and at child height (see Figure 1.).

In NZ, a proxy of socio-economic deprivation is school decile rating. Deciles are appointed to most schools, excluding some private schools, as a measure of the socio-economic position of the student community when compared to the rest of the country [38]. This project seeks to determine whether there are associations between school decile, distance from school, and walk and transit scores, and the prevalence of non-core food and beverage promotion at bus stops within walking distance from all Auckland schools.

## 2. Materials and Methods

This is a cross-sectional observational study, using Google Street View (GSV) [39] in order to assess advertising of food and beverages on bus shelters within a 500 m road network boundary of all schools in the Tāmaki Makaurau, Auckland region. Information, including authority, school type, decile, address [40], Walk Score^®^, and Transit Score [41], regarding all 573 schools in the region was entered into a Microsoft^®^ Excel^®^ spreadsheet and then imported into SPSS v.26 (IBM) for analysis.

### 2.1. School Decile Rating

School decile was measured as a proxy for socio-economic disadvantage in the area around the school. Ranging from one to ten, school decile ratings indicate the socioeconomic status of the neighbourhood students are from. The ten percent of schools with the greatest percentage of students from areas experiencing socioeconomic disadvantage is decile one [38]. The ten percent of schools with the lowest percentage of students from areas that experience socio-economic disadvantage is decile ten [38]. Deciles have been grouped into low (one-three), medium (four-seven), and high (seven-ten) for analysis in this study.

### 2.2. Walk Score^®^ and Transit Score

Walk Score^®^ and Transit Scores were measures as a proxy for walkability and public transport accessibility in the area around the school. Walk Score^®^ is a measure of the walkability of the area around the school that is based on the distance to nearby places and feasibility for pedestrian access [41]. Transit Score indicates the practicality of taking public transport to and from the school [41]. In the analysis of this study, both Walk Score^®^ and Transit Score are numbered from 1, most car dependent or minimal transport, to 5, least car dependent or abundant transport options.

### 2.3. Data Collection Protocol

The data collection protocol (See Appendix A) was informed by previous work [42], and an initial period of formative data collection and training of researchers (Authors 1, 2, and 3). Information that was collected from GSV image captures that was required for data entry included photo identification number, school name, bus stop address, distance from school boundary in metres, advertising categorisation, and GSV capture date. These data were recorded in Microsoft Excel. The protocol includes information on how to ‘travel’ the 500 m boundaries in GSV, identify bus stop advertisements, and code them according to core and non-core food and beverage types.

### 2.4. Researcher Training

The training of researchers on how to use GSV commenced before formal data collection. Comprehensive training on how to manoeuvre around streets, zoom, and identify outdoor advertisements on bus stops from different angles was included. Outdoor advertisements were defined as ‘stationary objects containing either a recognisable logo and/or an intended message’ [43]. Advertisements that were included were all bus shelter advertisements that were large enough to be seen on a 15-inch computer screen and with an identifiable logo or text. Advertisements on each side of double-sided bus shelters were identified and individually coded to ensure accurate data collection. Only the most recent image capture on GSV was used for this study.

After the identification of an outdoor advertisement at a bus stop, one screenshot for each advertisement was saved by school type and given a photo ID number to allow for data checking by the study manager (Author 6). The accurate identification of the closest street address for each outdoor advertisement was achieved by using the Map Tool in GSV. The shortest walking distance between the school and bus stop was calculated by the Distance Measurement Tool on GSV and then recorded in metres. The data collection protocol proved more detailed information (See Appendix A).

The inter-rater reliability testing was conducted between the two researchers, and formal data collection was not conducted until a minimum kappa value of 0.8 was reached [44]. Full primary and contributing state and state: integrated schools were assigned to one researcher (Author 3), and the remaining were assigned to the other (Author 1). Formal data collection was completed between August 2019 and January 2020. Following the data collection of each school type, the data were checked by the study manager (Author 6), who was available to assist researchers with categorisation of advertisements when uncertainties occurred. Figure 2 provides stages of data collection.

### 2.5. Coding Food and Beverage Advertisements

Bus stop advertisements for food and beverages were categorised in accordance with previous NZ-based advertising research [18]. The work of Signal et al. [18] is based on the WHO Regional Office for Europe Nutrient Profile Model [37]. Food or beverages that are recommended for marketing were coded “core” and not recommended were coded “non-core”. A combined category, “food and beverage”, was used when food and beverages were promoted together in on advertisement. Bus stop advertisements were assigned to the categories that are presented in Table 1.

### 2.6. Ground-Truthing

Following the completion of data collection for all schools, ground-truthing was completed in a random subset of schools. The RANDOM function in Microsoft Excel was used to identify a randomly selected ten per cent of schools. Ground-truthing involved checking for bus stops in person and comparing the data to the GSV data. There was 87.8% alignment between the ground-truthing results and the GSV data.

Upon completion of data collection and checking, the spreadsheet was cleaned and imported into IBM SPSS version 26 for analysis. All of the images have been saved on a secure drive at the University.

## 3. Results

The GSV images that were collected for this study were captured from February 2012 to May 2019, the average month of capture was July 2018. Of the 573 schools in the Auckland region, 190 had bus stop advertisements within a 500 m walking distance. This included 166 state or state: integrated schools out of 528 and 24 private: fully registered or private: provisionally registered schools out of 45. A total of 842 advertisements were identified, of which 702 were found around state or state: integrated schools and 140 were found around private: fully registered or private: provisionally registered schools.

The majority of advertisements found were coded “non-food other” (n = 541, 64.3%). 215 (25.5%) advertisements promoted food and/or beverage. More than half of all food and/or beverage advertisements found were coded “non-core” (n = 108, 50.2%). There were no advertisements promoting core food and beverages together. Table 2 presents descriptive results.

Examples of core food included yoghurt (n = 35, 42.7%) and breakfast cereals (n = 25, 30.5%), and the majority (n = 24, 96.0%) of core beverage advertisements were for full fat plain milk. Common non-core foods included chocolate bars (n = 27, 39.7%) and condiments, like tomato sauce and mayonnaise (n = 20, 29.4%). Examples of non-core beverages included coffee drinks (n = 14, 46.7%) and energy drinks (n = 5, 16.7%). All of the non-core food and beverage promotions were fast food (e.g., McDonald’s^®^).

### 3.1. School Decile

Advertising around low decile (one-three) schools contributed the greatest proportion of non-core food (n = 23, 33.8%), core food (n = 27, 32.9%), and non-core food and beverage advertisements (n = 5, 50.0%) identified, followed by high decile (eight-ten) schools (n = 19, 27.9%; n = 22, 26.8%; and, n = 3, 30.0% respectively). Overall, the greatest proportion of all non-core food and/or beverage advertisements were found near low decile schools (n = 36, 33.3%), followed by high decile (n = 31, 28.7%) and medium decile (n = 27, 25.0%) schools. Advertising around high decile (eight-ten) schools contributed the greatest proportion to core beverage advertisements found (n = 10, 40.0%), followed by medium decile (three-seven) schools (n = 8, 32.0%), see Table 3, below.

Some private schools have a decile rating, but most are not allocated one. Private schools without a decile rating were coded ‘N/A’. The socioeconomic status of the students who are more likely to attend private schools may be considered high because the typical annual school fee is around $20,000 [45]. When the high decile (eight-ten) schools and ‘N/A’ schools results are combined, the greatest proportion of all food and/or beverage advertisements categories were found near high socioeconomic areas (n = 91, 42.3%). The greatest proportion of all non-core advertisements will also be in this combined category (n = 45, 41.7%). However, this might be due to higher decile schools being more likely to be located in urban areas with many bus stops.

### 3.2. Distance from School Boundary

A greater number of total advertisements per 100 m were identified as the distance from the school increased. Food and/or beverage advertisements followed the same pattern (see Table 4 below).

### 3.3. Walk Score^®^

Schools with medium-high Walk Score^®^ (3 and 4) made up the greatest proportion of food and/or beverage advertising identified across all categories. The least number of advertisements found were near schools with the lowest Walk Score^®^ (1). There were no trend differences between core and non-core food and/or beverage advertisements and Walk Score^®^ (see Table 5, below).

The Pearson Chi-squared test was used to examine the relationship between the variables of combined variable core and non-core food and beverages and School decile (tertiles), Walk Score (quintiles), and Transit Score (quintiles). School decile was not statistically significant (Χ2 = 0.394, 2df, *p* = 0.821); the Walk Score was not statistically significant (Χ2 = 4.020, 4df, *p* = 0.403); and, Transit Score was not statistically significant (Χ2 = 2.189, 3df, *p* = 0.534). All of the analyses were carried out with SPSS v.26.

## 4. Discussion

The aim of this study was to quantify the amount of food and beverage that children are exposed to from bus stops around all schools in the Auckland region. The main findings of this research showed that 12.8% of all bus stop advertisements found were promoting non-core dietary options. The greatest proportion of all food and/or beverage advertisements were found near schools with high decile (eight-ten) ratings and private schools (n = 91, 42.3%). This might be due to higher decile schools being more likely to be located in urban areas with many bus stops. The number of bus stops with advertisements per school near high decile (eight-ten) and ‘N/A’ coded schools was 1.81, followed by 1.40 for low decile (one-three), and 1.11 for medium decile (four-seven) schools. Conversely, in a single decile category, the greatest proportion of non-core food and/or beverage advertisements were found near schools with low decile (one-three) ratings (n = 36, 33.3%). Furthermore, a greater proportion of advertisements were found as the distance from schools increased. Food and/or beverage advertisements also followed this trend. There were no significant trends regarding Walk Score^®^ and core and non-core advertisements.

### 4.1. Deprivation

There is a link between socio-economic deprivation and exposure to unhealthy food advertising in NZ and around the world. In this study, areas of higher deprivation with low decile (one-three) schools were where most of the non-core advertising was found. In another NZ study conducted by Vandevijvere et al. [15], shorter distances were reported between schools in areas of greater deprivation and unhealthy food outlets when compared to areas with lower deprivation. Similarly, in other countries, including Australia, Sweden, and the United Kingdom, people who live in suburbs that are more disadvantaged are more likely to be exposed to unhealthy food and beverage promotion [12,13,14]. Future research needs to be undertaken to confirm and study the relationship between socio-economic deprivation and the quantity of unhealthy food and beverage outdoor advertisements. It is important to remember, that bus stops are only one form of out-of-home that children are exposed to. This also includes billboards, street furniture, train stations, and on buses themselves. Besides schools, there are many other places that children frequent that could benefit from advertising restrictions to create healthier environments, such as parks, sport, and recreation facilities [21].

### 4.2. Strengths and Limitations

As with previous studies [42,46,47,48], this study also showed that Google Street View (GSV) is a useful and cost-effective tool for analysing neighbourhood characteristics. Screenshots can easily be taken, saved, and reviewed later. Neighbourhoods in a wide range of global contexts may also be assessed [49]. In comparison to field audits conducted on the ground, data collection in this way is safer and more cost and time effective [50,51]. A study found that a single researcher was able to audit more streets in half the time than four researchers were able to in the field [52]. This research is the first to ground-truth the use of GSV in comparison to on-the-ground field audits of outdoor advertising in NZ.

The limitations of this study included weaknesses of GSV imagery include quality, size, and the possibility of images being blocked or blurred [48]. The quality of some of the imagery that was acquired from GSV was not good enough to be able to distinguish what type of advertising was promoted. Some advertisements featured large images of people’s faces, which were blurred for privacy and hid the identifying feature of an advertisement. Another limitation was that Google would not have all of the entrances and exits of a school mapped; therefore, there might be other bus stops within 500 m walking distance that were unable to be included in the data collection. Moreover, some roads or parts of roads have not been recorded in street view, so some bus stops may have been missed. There were obstructions by people or large vehicles that made it difficult to code the advertisements or see if there were any. Although the average month of GSV capture was July 2018, some most recent street view captures for a particular area may date as far back as 2012. This presents a limitation to using GSV to assess and monitor change with policy interventions, particularly in areas where GSV data are not regularly updated.

A further limitation of the study design was that only bus stops were included for data collection due to time and resource constraints. The quantity of unhealthy food and beverage advertisements seen by children identified in this study is likely to be conservative in comparison to the amount that they are really exposed to. There are future research opportunities for collecting data on all the advertisements seen around schools or even the whole Auckland region. Another limitation was that there was overlap in some school boundaries mapped, so the quantity of bus stop advertisements found is not absolute.

### 4.3. Implications for Policy and Practice

In many developed countries, self-regulation is the main policy response to controlling the advertisements that children are exposed to [53]. Since the implementation of self-regulated advertising codes, several studies have shown that there have been no significant changes in the exposure of children to unhealthy food and beverage advertisements [54,55,56]. However, many countries are shifting towards having more restrictions being imposed on these codes [53]. Though the need and interest in implementing increasingly restrictive policies is evident [31,32,33], barriers to these policy changes exist [32,33,53,57].

The self-regulated Advertising Standards Authority (ASA) [35] controls what children see in advertisements in NZ. The first rule in the children and young people section of the code states that “advertisements for occasional food and beverage must not target children”. However, there were many advertisements around schools that promoted non-core food and beverages, as identified from our data collection. Advertisements in places where children frequent are considered to target children, according to the World Health Organisation [58] and ASA [35]. An example, given by ASA, is schools [35], but this should be extended to the areas around schools, as children are likely to pass through those areas twice every weekday.

As loopholes are being exploited when it comes to advertising to children [54,55], the execution of the ASA code [35] requires review. The weakness of the current self-regulatory code emphasises the necessity for more government participation to amend the framework for advertising to children, including travel routes to and from school. Recommendations made by the World Health Organisation [37,59] should be reinforced in order to decrease exposure of unhealthy marketing to children. Increased determination to ensure that children are protected from the effects of unhealthy dietary marketing should be made [59], as they are unlikely to understand their persuasive intentions [27,28,29].

The weakness of the current self-regulatory code emphasises the necessity for government regulations on unhealthy food and beverage marketing to children, including travel routes to and from school. However in NZ, local governments lack powers to impose further restrictions. However, this study highlights how local governments could explore their existing powers to implement restrictions, for example, preventing unhealthy marketing from being placed on government assets e.g., bus stops, particularly those that are located in areas where children frequently gather.

## 5. Conclusion

This research is the first to map all bus stops around schools within a large, urban city and presents the first comprehensive analysis of bus stop advertising around schools in Tāmaki Makaurau, Auckland. 12.8% of all bus stop advertisements that are found in this study promoted non-core dietary options. Therefore, it would be a small change, in terms of policy and advertising revenue, if non-core dietary options were no longer permitted to be advertised near schools. Quantitative evidence is provided for advocacy to changes in national advertising policies and marketing standards in NZ. This novel research highlights the utility of Google Street View for large city-scale audits and details evidence of the obesogenic advertising children are exposed to regularly on their way to and from school.

## Figures and Tables

**Figure 1 nutrients-12-01192-f001:**
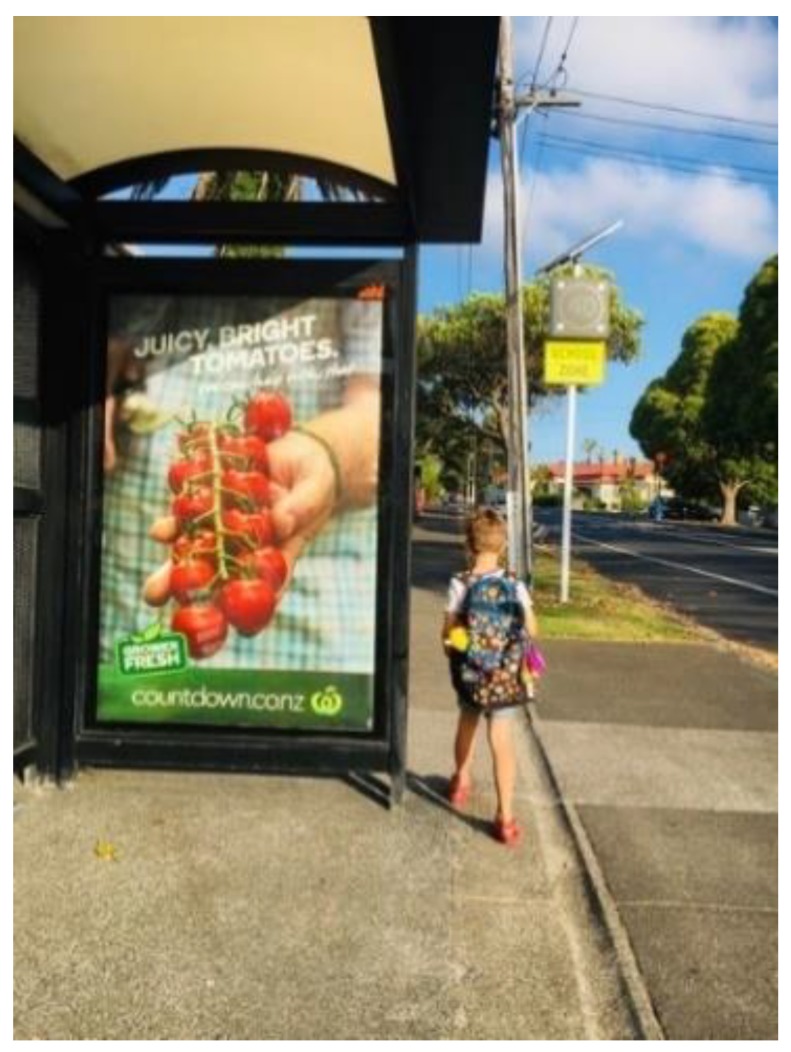
Child standing next to a typical bus stop.

**Figure 2 nutrients-12-01192-f002:**
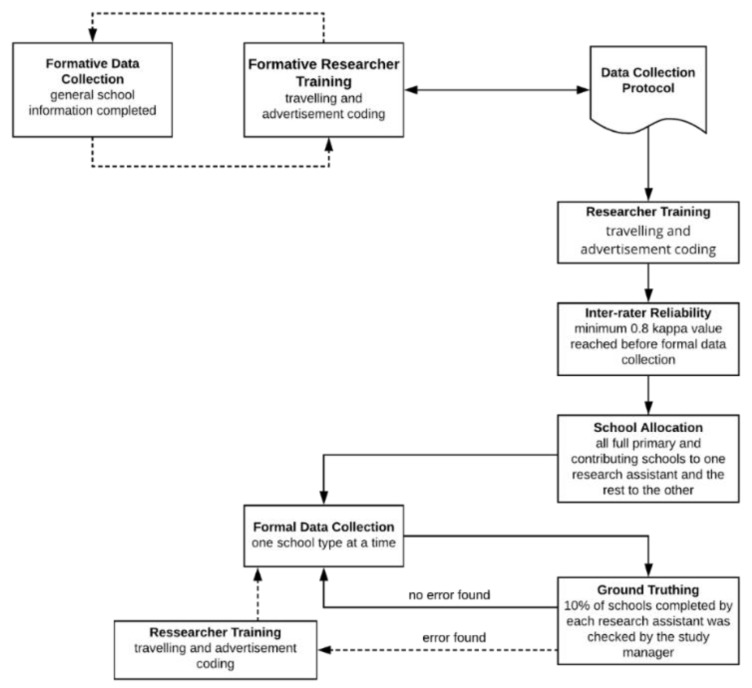
Flow Chart of Data Collection Processes.

**Figure A1 nutrients-12-01192-f0A1:**
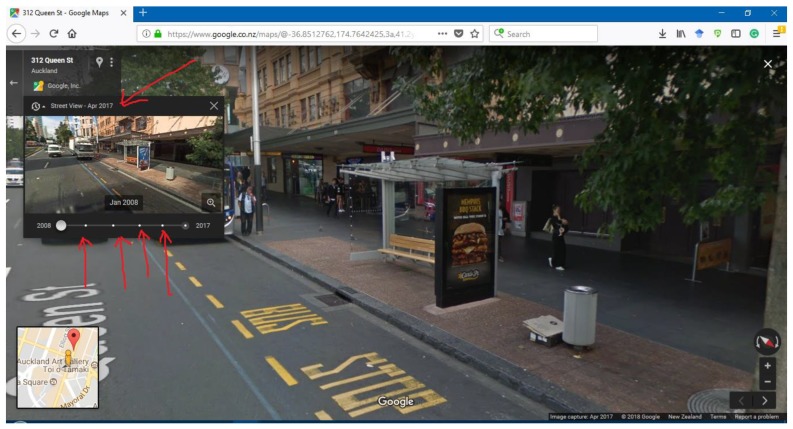
Image Capture Date Example.

**Figure A2 nutrients-12-01192-f0A2:**
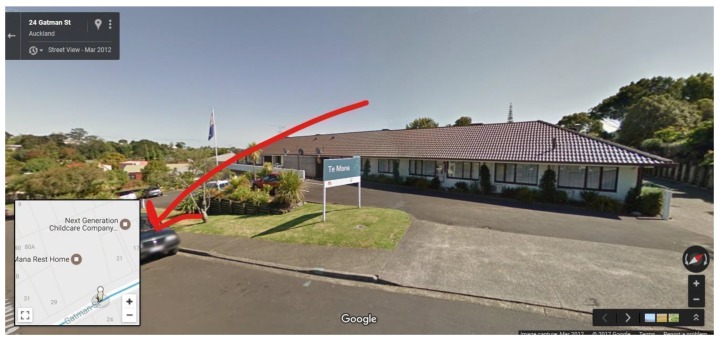
Map Tool Example.

**Figure A3 nutrients-12-01192-f0A3:**
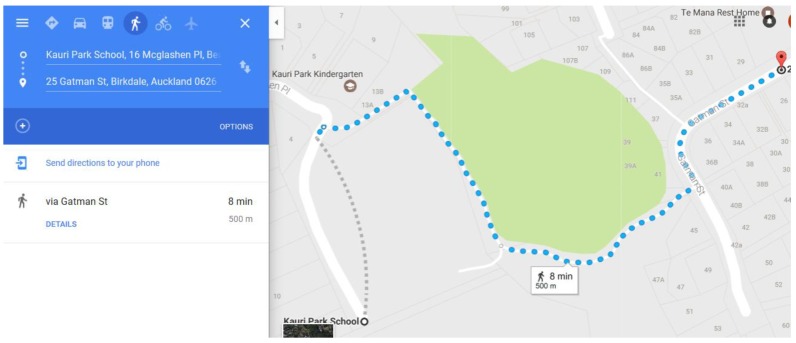
Distance on Foot Example.

**Table 1 nutrients-12-01192-t001:** Advertisements Identified by Category.

Advertisement Category	Examples
Food Non-core	Chocolate, fast food, ice cream
Food Core	Yoghurt, bread, vegetables
Beverage Non-core	Flavoured coffee, energy drinks, juice
Beverage Core	Milk, water, reduced sugar juice
Food and Beverage Non-core	Fast food combos: soft drink and burger
Food and Beverage Core	Non-flavoured milk and bread
Non-food other	Services: mobile service provider, clothing
Non-food residential	Real estate: rent, buy, open home
Unable to Categorise—image unclear/blocked/angle	N/A

**Table 2 nutrients-12-01192-t002:** Basic Descriptive Table of Results.

		Number of Advertisements n (%)
Distance from School (m)	≤100	92 (10.9)
	101–200	130 (15.4)
	201–300	160 (19.0)
	301–400	211 (25.1)
	401–500	249 (29.6)
Decile	Low (1–3)	285 (33.8)
	Medium (4–7)	178 (21.1)
	High (8–10)	311 (36.9)
	N/A	68 (8.1)
Walk Score	1 ^1^	12 (1.4)
	2	162 (19.2)
	3	282 (33.5)
	4	314 (37.3)
	5 ^2^	72 (8.6)
Transit Score	1 ^3^	1 (0.1)
	2	370 (43.9)
	3	352 (41.8)
	4	45 (5.3)
	5 ^4^	58 (6.9)
	N/A	16 (1.9)
Advertisement Code	Food Non-core	68 (8.1)
	Food Core	82 (9.7)
	Beverage Non-core	30 (3.6)
	Beverage Core	25 (3.0)
	Food & Beverage Non-core	10 (1.2)
	Non-food Other	541 (64.3)
	Non-food Residential	30 (3.6)
	Unable to Categorise	56 (6.7)

^1^ lowest walkability ^2^ highest walkability ^3^ poorest transport options ^4^ greatest transport options.

**Table 3 nutrients-12-01192-t003:** Proportion of Food and Beverage Advertisements by School Decile.

		Advertisement Code n (%)					
		Food Non-Core	Food Core	Beverage Non-Core	Beverage Core	Food & Beverage Non-Core	Total
**Decile**	**Low (1–3)**	23 (33.8)	27 (32.9)	8 (26.7)	5 (20.0)	5 (50.0)	68 (31.6)
	**Medium (4–7)**	16 (23.5)	21 (25.6)	9 (30.0)	8 (32.0)	2 (20.0)	56 (26.0)
	**High (8–10)**	19 (27.9)	22 (26.8)	9 (30.0)	10 (40.0)	3 (30.0)	63 (29.3)
	**N/A**	10 (14.7)	12 (14.6)	4 (13.3)	2 (8.0)	0 (0.0)	28 (13.0)
**Total**		68 (100.0)	82 (100.0)	30 (100.0)	25 (100.0)	10 (100.0)	215 (100.0)

**Table 4 nutrients-12-01192-t004:** Proportion of Advertisements by Distance from School Boundary.

		Advertisement Code n (%)								
		Food Non-Core	Food Core	Beverage Non-Core	Beverage Core	Food & Beverage Non-Core	Non-Food Other	Non-Food Residential	Unable to Categorise	Total
**Distance from School Boundary (metres)**	**≤100**	6 (8.8)	14 (17.1)	0 (0.0)	3 (12.0)	0 (0.0)	60 (11.1)	7 (23.3)	2 (3.6)	92 (10.9)
	**101–200**	8 (11.8)	13 (15.9)	3 (10.0)	6 (24.0)	1 (10.0)	84 (15.5)	4 (13.3)	11 (19.6)	130 (15.4)
	**201–300**	10 (14.7)	11 (13.4)	1 (3.3)	2 (8.0)	1 (10.0)	115 (21.3)	6 (20.0)	14 (25.0)	160 (19.0)
	**301–400**	14 (20.6)	21 (25.6)	14 (46.7)	7 (28.0)	2 (20.0)	138 (25.5)	3 (10.0)	12 (21.4)	211 (25.1)
	**401–500**	30 (44.1)	23 (28.0)	12 (40.0)	7 (28.0)	6 (60.0)	144 (26.6)	10 (33.3)	17 (30.4)	249 (29.6)
**Total**		68 (100.0)	82 (100.0)	30 (100.0)	25 (100.0)	10 (100.0)	541 (100.0)	30 (100.0)	56 (100.0)	842 (100.0)

**Table 5 nutrients-12-01192-t005:** Proportion of Food and Beverage Advertisements by Walk Score *.

	Advertisement Code n (%)						
		Food Non-Core	Food Core	Beverage Non-Core	Beverage Core	Food & Beverage Non-Core	Total
Walk Score^®^	1	0 (0.0)	2 (2.4)	0 (0.0)	1 (4.0)	0 (0.0)	3 (1.4)
	2	12 (17.6)	9 (11.0)	5 (16.7)	3 (12.0)	1 (10.0)	30 (14.0)
	3	24 (35.3)	23 (28.0)	9 (30.0)	8 (32.0)	2 (20.0)	66 (30.7)
	4	21 (30.9)	34 (41.5)	13 (43.3)	11 (44.0)	7 (70.0)	86 (40.0)
	5	11 (16.1)	14 (17.1)	3 (10.0)	2 (8.0)	0 (0.0)	30 (14.0)
Total		68 (100.0)	82 (100.0)	30 (100.0)	25 (100.0)	10 (100.0)	215 (100.0)

* Walk Score^®^ is a measure of the walkability of the area around the school based on the distance to nearby places and feasibility for pedestrian access [41]. For this study, Walk Score^®^ is numbered from 1, most car dependent, to 5, least car dependent.

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
