# Peer review of "Bus Stops Near Schools Advertising Junk Food and Sugary Drinks"

_nutrients, 2020, doi:10.3390/nu12041192_

Round 1

Reviewer 1 Report

This manuscript aims to present the results from a very original study on children’s food environment in Auckland. The aim of the study is to explore the proportion of bus stop advertisements promoting non-core food and 22 beverages within walking distance from schools in Tāmaki Makaurau using Google Street View.

The analysis of children’s food environment through bus stops advertisement is very interesting and of great public health importance. However, some weaknesses in the study would deserve more attention. The main weakness of this study are the statistical plan and analysis methods used.

  1. Introduction/description of the intervention:
  1. The paper would benefit from providing more information in lines 52-59, by being more explicit and give percentages, particularly on prevalence of overweight and obesity and proportion male and female from the indigenous population living in deprived areas. Numbers accompanying the statements would be welcomed.
  2. More references in background, existing literature and frameworks should be added (for example, see Unicef-Innocenti framework for children’s food environment).
  1. Statistical analysis:
    1. The statistical analysis is limited and too descriptive. Only conducting one test, using chi2, is limited and not the more adapted test, because of the too many sub-groups and small samples in some cells.
    2. Have the researchers performed comparisons between non-core Food and Core Food? Between non-core beverages and core beverages? This study could benefit from a better statistical analysis plan.
  2. Results
    1. The level of evidence of the study is weak because of the weakness of the statistical analysis plan. This can be improved.
  3. Formatting of tables:
    1. All tables need to be checked and edited to improve the visual (broken lines…)
    2. Add footnotes, to indicate labels on score variables (for example in table 2: low walkability, high walkability)

This study collected very interesting data and has an interesting and relevant research question. The paper would benefit from the support of a statistician to modify and improve the statistical plan for analysis and provide better quality results.

Author Response

Thank you very much for your valuable comments.

Many thanks for the time taken by the reviewers and editor to provide constructive feedback on this manuscript. We have made changes as recommended by the reviewers and these are reflected in track changes in the updated manuscript. We believe the feedback and corresponding changes have strengthened the manuscript and look forward to hearing from you in due course.

Reviewer 1

Author

This manuscript aims to present the results from a very original study on children’s food environment in Auckland. The aim of the study is to explore the proportion of bus stop advertisements promoting non-core food and 22 beverages within walking distance from schools in Tāmaki Makaurau using Google Street View.

The analysis of children’s food environment through bus stops advertisement is very interesting and of great public health importance. However, some weaknesses in the study would deserve more attention. The main weakness of this study are the statistical plan and analysis methods used.

Thank you for your review of this manuscript. We have made changes in response to the comments as noted below and hope these are to the satisfaction of the reviewer.

Introduction/description of the intervention:

The paper would benefit from providing more information in lines 52-59, by being more explicit and give percentages, particularly on prevalence of overweight and obesity and proportion male and female from the indigenous population living in deprived areas. Numbers accompanying the statements would be welcomed.

From lines 52-65, percentages and n values have been given where possible regarding prevalence of obesity in children living in the most deprived areas, comparison of Māori and non-Māori living in the most deprived areas, and the proportion of male and female from the indigenous population living in the most deprived areas.

Many inequities regarding obesity prevalence among children exist. In Aotearoa, New Zealand (hereafter, NZ), and around the world, people who live in the most socioeconomically deprived neighbourhoods are more likely to be affected by excess body size [9, 10]. When compared to children living in the least socioeconomically deprived areas, children living in the most deprived areas were 1.6 times as likely to be obese [11]. There is lower access to healthy foods [12] and greater exposure to unhealthy foods [12-15] in more deprived neighbourhoods. In NZ, Māori (New Zealand’s Indigenous population) are more likely to live in the most deprived areas (n = 140,886, 23.5%) compared to Non-Māori (n = 232,779, 6.8%) [11]. Females (n = 75,117, 24.3%) are also more likely to live in the most deprived areas compared to males (n = 65,712, 22.8%) [11]. Children of Māori (28.4%)  and Pacific Island (15.5%) ethnicity have considerably higher prevalence of obesity compared to New Zealand European/Other ethnicities (8.2%) [9].

More references in background, existing literature and frameworks should be added (for example, see Unicef-Innocenti framework for children’s food environment).

The Innocenti framework, and the socio-ecological model of health behaviours, and accompanying references, have been added to the introduction as follows.

 This does not align with the Convention on the Rights of the Child [19]. The presence of unhealthy food and drink advertisements in areas children visit every day impacts upon children’s food choices [20, 21], this is highlighted in the UNICEF-Innocenti framework on food systems for children and adolescents [22]. Outside of the family setting, school is the most important environment for children and adolescents [23-25]. This research is underpinned by the socio-ecological model of health behaviours [26].

Statistical analysis:

The statistical analysis is limited and too descriptive. Only conducting one test, using chi2, is limited and not the more adapted test, because of the too many sub-groups and small samples in some cells.

We conducted the statistical analysis with the guidance and support of a biostatistician, who is thanked in the Acknowledgements section of the manuscript. We explored thoroughly all statistical options for analysis and upon the guidance and advice of the biostatistician, we provide these descriptive results that best address the overall aim of this study, which is to: “explore the proportion of bus stop advertisements promoting non-core (not recommended by WHO) food and beverages within walking distance (500m) from schools in Auckland, NZ.”

Further, this research used an emerging method (GSV) with multiple researchers responsible for data collection. As such a kappa co-efficient was conducted for inter-rater reliability testing (see lines 158-164).

Overall, this paper presents important information that is relevant and easily understood by researchers, practitioners and policy makers alike. Researchers will be able to build upon the foundations laid in this study to develop future research projects that include more complex statistical analysis plans.

Have the researchers performed comparisons between non-core Food and Core Food? Between non-core beverages and core beverages? This study could benefit from a better statistical analysis plan.

The results of comparisons between non-core and core food are presented in lines 196 -199. 

The majority of advertisements found were coded “non-food other” (n = 541, 64.3%). 215 (25.5%) advertisements promoted food and/or beverage. More than half of all food and/or beverage advertisements found were coded “non-core” (n = 108, 50.2%).

Lines 257 onwards state: The Pearson Chi-squared test was used to examine the relationship between the variables of combined variable core and non-core food and beverages and School decile (tertiles), Walk Score (quintiles), and Transit Score (quintiles). School decile was not statistically significant (Χ2= .394, 2df, p = 0.821); Walk Score was not statistically significant (Χ2=4.020,4df, p=0.403); Transit Score was not statistically significant (Χ2= 2.189, 3df, p=0.534).

Results

The level of evidence of the study is weak because of the weakness of the statistical analysis plan. This can be improved.

Our analysis plan achieves the aim of the study.

Formatting of tables:

All tables need to be checked and edited to improve the visual (broken lines…)

The tables are currently formatted as per journal requirements. If/when this manuscript gets to the production stage, we will assist the production team to amend the broken line issue, as much as possible.

Add footnotes, to indicate labels on score variables (for example in table 2: low walkability, high walkability)

Foot notes have been added to table 2 and lines 209-212.

1 lowest walkability

2 highest walkability

3 poorest transport options

4 greatest transport options

This study collected very interesting data and has an interesting and relevant research question. The paper would benefit from the support of a statistician to modify and improve the statistical plan for analysis and provide better quality results.

A biostatistician was consulted throughout the research and is thanked in the acknowledgements. Our analysis plan achieves the aim of the study.

Reviewer 2

Author

Minor clarifications:

Line 153-154 – Data collection completed Aug 2019-Jan 2020, however, on lines 184-5, GSV data was from as far back as 2012 (median 2018). While I understand that the researchers have no control over when GSV data were captured, older captures may not reflect the situation at the time of other data collection. This needs to be addressed in the limitations. Alternatively, particularly out of date GSV captures could be removed from the data base. this also poses a limitation to ongoing monitoring to assess changes with policy interventions

The following lines have been added at the end of the paragraph at line 321.

Although the average month of GSV capture was July 2018, some most recent street view captures for a particular area may date as far back as 2012. This presents a limitation to using GSV to assess and monitor change with policy interventions, particularly in areas where GSV data is not regularly updated.

Reviewer 3

Author

This is an important study that can change the policy related to the advertisments targeting the children. The Promotion of the inhealthy food and beverages is a serious problem that can change in a long run the dietary habits.  

Thank you so much for taking the time to review this research.

We agree that this is an important work that will have direct influence on policy to change the food environment in which children live in.

Reviewer 2 Report

This is an interesting manuscript and an innovative methodology.

Minor clarifications:

Line 153-154 – Data collection completed Aug 2019-Jan 2020, however, on lines 184-5, GSV data was from as far back as 2012 (median 2018). While I understand that the researchers have no control over when GSV data were captured, older captures may not reflect the situation at the time of other data collection. This needs to be addressed in the limitations. Alternatively, particularly out of date GSV captures could be removed from the data base. this also poses a limitation to ongoing monitoring to assess changes with policy interventions

Author Response

Many thanks for the time taken by the reviewers and editor to provide constructive feedback on this manuscript. We have made changes as recommended by the reviewers and these are reflected in track changes in the updated manuscript. We believe the feedback and corresponding changes have strengthened the manuscript and look forward to hearing from you in due course.

Reviewer 1

Author

This manuscript aims to present the results from a very original study on children’s food environment in Auckland. The aim of the study is to explore the proportion of bus stop advertisements promoting non-core food and 22 beverages within walking distance from schools in Tāmaki Makaurau using Google Street View.

The analysis of children’s food environment through bus stops advertisement is very interesting and of great public health importance. However, some weaknesses in the study would deserve more attention. The main weakness of this study are the statistical plan and analysis methods used.

Thank you for your review of this manuscript. We have made changes in response to the comments as noted below and hope these are to the satisfaction of the reviewer.

Introduction/description of the intervention:

The paper would benefit from providing more information in lines 52-59, by being more explicit and give percentages, particularly on prevalence of overweight and obesity and proportion male and female from the indigenous population living in deprived areas. Numbers accompanying the statements would be welcomed.

From lines 52-65, percentages and n values have been given where possible regarding prevalence of obesity in children living in the most deprived areas, comparison of Māori and non-Māori living in the most deprived areas, and the proportion of male and female from the indigenous population living in the most deprived areas.

Many inequities regarding obesity prevalence among children exist. In Aotearoa, New Zealand (hereafter, NZ), and around the world, people who live in the most socioeconomically deprived neighbourhoods are more likely to be affected by excess body size [9, 10]. When compared to children living in the least socioeconomically deprived areas, children living in the most deprived areas were 1.6 times as likely to be obese [11]. There is lower access to healthy foods [12] and greater exposure to unhealthy foods [12-15] in more deprived neighbourhoods. In NZ, Māori (New Zealand’s Indigenous population) are more likely to live in the most deprived areas (n = 140,886, 23.5%) compared to Non-Māori (n = 232,779, 6.8%) [11]. Females (n = 75,117, 24.3%) are also more likely to live in the most deprived areas compared to males (n = 65,712, 22.8%) [11]. Children of Māori (28.4%)  and Pacific Island (15.5%) ethnicity have considerably higher prevalence of obesity compared to New Zealand European/Other ethnicities (8.2%) [9].

More references in background, existing literature and frameworks should be added (for example, see Unicef-Innocenti framework for children’s food environment).

The Innocenti framework, and the socio-ecological model of health behaviours, and accompanying references, have been added to the introduction as follows.

 This does not align with the Convention on the Rights of the Child [19]. The presence of unhealthy food and drink advertisements in areas children visit every day impacts upon children’s food choices [20, 21], this is highlighted in the UNICEF-Innocenti framework on food systems for children and adolescents [22]. Outside of the family setting, school is the most important environment for children and adolescents [23-25]. This research is underpinned by the socio-ecological model of health behaviours [26].

Statistical analysis:

The statistical analysis is limited and too descriptive. Only conducting one test, using chi2, is limited and not the more adapted test, because of the too many sub-groups and small samples in some cells.

We conducted the statistical analysis with the guidance and support of a biostatistician, who is thanked in the Acknowledgements section of the manuscript. We explored thoroughly all statistical options for analysis and upon the guidance and advice of the biostatistician, we provide these descriptive results that best address the overall aim of this study, which is to: “explore the proportion of bus stop advertisements promoting non-core (not recommended by WHO) food and beverages within walking distance (500m) from schools in Auckland, NZ.”

Further, this research used an emerging method (GSV) with multiple researchers responsible for data collection. As such a kappa co-efficient was conducted for inter-rater reliability testing (see lines 158-164).

Overall, this paper presents important information that is relevant and easily understood by researchers, practitioners and policy makers alike. Researchers will be able to build upon the foundations laid in this study to develop future research projects that include more complex statistical analysis plans.

Have the researchers performed comparisons between non-core Food and Core Food? Between non-core beverages and core beverages? This study could benefit from a better statistical analysis plan.

The results of comparisons between non-core and core food are presented in lines 196 -199. 

The majority of advertisements found were coded “non-food other” (n = 541, 64.3%). 215 (25.5%) advertisements promoted food and/or beverage. More than half of all food and/or beverage advertisements found were coded “non-core” (n = 108, 50.2%).

Lines 257 onwards state: The Pearson Chi-squared test was used to examine the relationship between the variables of combined variable core and non-core food and beverages and School decile (tertiles), Walk Score (quintiles), and Transit Score (quintiles). School decile was not statistically significant (Χ2= .394, 2df, p = 0.821); Walk Score was not statistically significant (Χ2=4.020,4df, p=0.403); Transit Score was not statistically significant (Χ2= 2.189, 3df, p=0.534).

Results

The level of evidence of the study is weak because of the weakness of the statistical analysis plan. This can be improved.

Our analysis plan achieves the aim of the study.

Formatting of tables:

All tables need to be checked and edited to improve the visual (broken lines…)

The tables are currently formatted as per journal requirements. If/when this manuscript gets to the production stage, we will assist the production team to amend the broken line issue, as much as possible.

Add footnotes, to indicate labels on score variables (for example in table 2: low walkability, high walkability)

Foot notes have been added to table 2 and lines 209-212.

1 lowest walkability

2 highest walkability

3 poorest transport options

4 greatest transport options

This study collected very interesting data and has an interesting and relevant research question. The paper would benefit from the support of a statistician to modify and improve the statistical plan for analysis and provide better quality results.

A biostatistician was consulted throughout the research and is thanked in the acknowledgements. Our analysis plan achieves the aim of the study.

Reviewer 2

Author

Minor clarifications:

Line 153-154 – Data collection completed Aug 2019-Jan 2020, however, on lines 184-5, GSV data was from as far back as 2012 (median 2018). While I understand that the researchers have no control over when GSV data were captured, older captures may not reflect the situation at the time of other data collection. This needs to be addressed in the limitations. Alternatively, particularly out of date GSV captures could be removed from the data base. this also poses a limitation to ongoing monitoring to assess changes with policy interventions

The following lines have been added at the end of the paragraph at line 321.

Although the average month of GSV capture was July 2018, some most recent street view captures for a particular area may date as far back as 2012. This presents a limitation to using GSV to assess and monitor change with policy interventions, particularly in areas where GSV data is not regularly updated.

Reviewer 3

Author

This is an important study that can change the policy related to the advertisments targeting the children. The Promotion of the inhealthy food and beverages is a serious problem that can change in a long run the dietary habits.  

Thank you so much for taking the time to review this research.

We agree that this is an important work that will have direct influence on policy to change the food environment in which children live in.

Reviewer 3 Report

This is an important study that can change the policy related to the advertisments targeting the children. The Promotion of the inhealthy food and beverages is a serious problem that can change in a long run the dietary habits.  

Author Response

(The authors gave the same response as above.)

Round 2

Reviewer 1 Report

The authors have provided a new version and a satisfactory reply to my comments. The manuscript has been significantly improved and is ready for publication in Nutrients.